**Data Availability Statement:** The data contain potentially identifying or sensitive patient

# Incidence and risk factors for major bleeding among patients undergoing percutaneous coronary intervention: Findings from the Norwegian Coronary Stent Trial (NORSTENT)

**Per-Jostein Samuelsen**[ORCID][1,2]*, **Anne Elise Eggen**[2], **Terje Steigen**[3,4], **Tom Wilsgaard**[2], **Andreas Kristensen**[3], **Anne Skogsholm**[3], **Elizabeth Holme**[3], **Christian van den Heuvel**[3], **Jan Erik Nordrehaug**[5], **Bjørn Bendz**[6,7], **Dennis W. T. Nilsen**[5,8], **Kaare Harald Bønaa**[ORCID][9,10]

1 Regional Medicines Information and Pharmacovigilance Centre (RELIS), University Hospital of North Norway, Tromsø, Norway, 2 Department of Community Medicine, UiT The Arctic University of Norway, Tromsø, Norway, 3 Department of Cardiology, University Hospital of North Norway, Tromsø, Norway, 4 Cardiovascular Diseases Research Group, UiT The Arctic University of Norway, Tromsø, Norway, 5 Department of Clinical Science, University of Bergen, Bergen, Norway, 6 Department of Cardiology, Rikshospitalet, Oslo University Hospital, Oslo, Norway, 7 Faculty of Medicine, Institute of Clinical Medicine, University of Oslo, Oslo, Norway, 8 Department of Cardiology, Stavanger University Hospital, Stavanger, Norway, 9 Clinic for Heart Disease, St. Olav's University Hospital, Trondheim, Norway, 10 Department of Circulation and Medical Imaging, Norwegian University of Science and Technology, Trondheim, Norway

* per-jostein.samuelsen@unn.no

## Abstract

### Introduction

Bleeding is a concern after percutaneous coronary intervention (PCI) and subsequent dual antiplatelet therapy (DAPT). We herein report the incidence and risk factors for major bleeding in the Norwegian Coronary Stent Trial (NORSTENT).

### Materials and methods

NORSTENT was a randomized, double blind, pragmatic trial among patients with acute coronary syndrome or stable coronary disease undergoing PCI during 2008–11. The patients (N = 9,013) were randomized to receive either a drug-eluting stent or a bare-metal stent, and were treated with at least nine months of DAPT. The patients were followed for a median of five years, with Bleeding Academic Research Consortium (BARC) 3–5 major bleeding as one of the safety endpoints. We estimated cumulative incidence of major bleeding by a competing risks model and risk factors through cause-specific Cox models.

### Results

The 12-month cumulative incidence of major bleeding was 2.3%. Independent risk factors for major bleeding were chronic kidney disease, low bodyweight (< 60 kilograms), diabetes mellitus, and advanced age (> 80 years). A myocardial infarction (MI) or PCI during follow-up increased the risk of major bleeding (HR = 1.67, 95% CI 1-29-2.15).

information. Data cannot be shared publicly because of restriction imposed by the Regional ethics committee (post@helseforskning.etikkom. no) and through the Norwegian data protection legislation. Data can be made available after approval by the NORSTENT executive steering committee (coordinating center: Department of Community Medicine, UiT The Arctic University of Norway, postmottak@uit.no), for researchers who meet the criteria for access to confidential data.

**Funding:** This work was funded by a grant to PJS from the Northern Norway Regional Health Authority (Helse Nord RHF; https://helse-nord.no/), grant no. HNF1420-18. NORSTENT was funded by Norwegian Research Council and others. The funders had no role in study design, data collection and analysis, decision to publish, or preparation of the manuscript.

**Competing interests:** The authors have declared that no competing interests exist.

## Conclusions

The 12-month cumulative incidence of major bleeding in NORSTENT was higher than reported in previous, explanatory trials. This analysis strengthens the role of chronic kidney disease, advanced age, and low bodyweight as risk factors for major bleeding among patients receiving DAPT after PCI. The presence of diabetes mellitus or recurrent MI among patients is furthermore a signal of increased bleeding risk.

## Clinical trial registration

Unique identifier NCT00811772; http://www.clinicaltrial.gov.

## 1 Introduction

Dual antiplatelet therapy (DAPT) is mandatory after percutaneous coronary intervention (PCI) with stent implantation in order to prevent ischemic events [1]. The optimal duration of DAPT, however, is debated [2]. The risk of ischemic events must be balanced against the risk of major bleeding imposed by the antithrombotic therapy. For this purpose, several risk prediction tools based on randomized controlled trials (RCTs) [3,4] or observational studies [5], have been developed. Of note, recent European guidelines [1] and other guidelines worldwide [6–8] recommend or suggest the use of standardized risk scores, e.g., the DAPT and PRECISE-DAPT scores, to optimize the duration of DAPT. However, the validity and clinical utility of such scores have been questioned [9]. Obviously, an assessment of ischemic and bleeding risk relies on valid data. In brief, observational studies, and in particular registry-based studies, often have a high external validity but have methodological challenges, such as the lack of validated endpoints, and confounding. On the other hand, explanatory RCTs often have highly selected study populations that do not necessarily represent the clinical populations in question. Pragmatic RCTs, however, have less strict inclusion and exclusion criteria and produce study populations that more closely resemble clinical practice, still offering the advantage of increased validity through adjudicated clinical endpoints.

The aims of the current analysis were: 1) to estimate the incidence of major bleeding and to identify subgroups with increased bleeding risks among stent naïve patients undergoing PCI with stent implantation and DAPT, 2) to identify risk factors for major bleeding, including risks associated with a new or recurrent MI or PCI during follow-up. This was achieved using data from the Norwegian Coronary Stent Trial (NORSTENT) [10].

## 2 Materials and methods

### 2.1 Study population

NORSTENT was a randomized, controlled, open-label, multicenter, pragmatic trial [10]. Eligible participants were stent naïve patients ≥ 18 years with acute coronary syndrome (i.e., MI or unstable angina pectoris) or stable coronary artery disease who were candidates for percutaneous coronary intervention (PCI) with stent implantation. Patients with contraindications towards DAPT, or patients on oral anticoagulants were excluded. Patients were recruited between 2008 and 2011, with a maximum follow-up until December 31, 2014.

The participants were randomized to receive either drug-eluting stent(s) (DES) or bare-metal stent(s) (BMS). Patients were prescribed DAPT with acetylsalicylic acid (ASA) 75 mg daily indefinitely and clopidogrel 75 mg daily for at least nine months after the PCI, regardless

of the stent type. Other cardiovascular drugs, e.g., statins, were given according to contemporary clinical guidelines and practice.

## 2.2 Endpoint

The endpoint was major bleeding, defined as a Bleeding Academic Research Criteria (BARC) bleeding of 3, 4, or 5 [11]. This endpoint included overt bleeding plus hemoglobin drop of 3 to < 5 g/dL, overt bleeding plus hemoglobin drop ≥ 5 g/dL, cardiac tamponade, bleeding requiring surgical intervention, or intravenous vasoactive agents, intracranial or intraocular bleeding, major bleeding related to coronary artery by-pass surgery, or fatal bleeding. Endpoints were collected by means of electronic linkage to the Norwegian Patient Registry with the use of a unique 11-digit Norwegian national identification number for each patient. The patient registry includes the codes of The International Classification of Diseases, 10th Revision (ICD-10), with respect to all the main diagnoses and up to 20 secondary diagnoses and all procedure codes from all hospitalizations in Norway. A broad search was made to identify any hospitalization for cardiovascular disease and any hospitalization for suspected bleeding. The specific search criteria have been published [10]. All outcomes were adjudicated by an endpoint committee based on information in hospital records.

## 2.3 Statistical analysis

We treated the entire study population as a single cohort, i.e., both study arms were merged.

**2.3.1 Definition of variables.** Chronic kidney disease was defined as an eGFR < 60 ml/min/1.73 $m^2$ (CKD-EPI, assuming an all-Caucasian study population). Low bodyweight was defined as body weight < 60 kilograms, or underweight as a body mass index < 18.5 kg/$m^2$ in sensitivity analyses. Hypertension, hyperlipidemia, or diabetes mellitus were defined to be present if the patient received pharmacological treatment for any of these conditions. HbA1C was also measured in a subgroup of $n$ = 5,267, and diabetes mellitus defined as HbA1C ≥ 6.5% was also used in sensitivity analyses. Acute coronary syndrome (ACS) was defined as non-ST-elevation MI, ST-elevation MI, or unstable angina pectoris. The stent length refers to the total stent length per patient, i.e., a patient could receive several stents in one or multiple lesions.

**2.3.2 Multiple imputation.** Among all variables considered (Table 1), the combined proportion of missing observations was 17% with a possible missing at random pattern [12,13]. We therefore used multiple imputation with substantive model compatible fully conditional specification [14]. We recoded or transformed the variables before the imputation [12,13], e.g., chronic kidney disease, and included all variables in our imputation model, and ran 20 imputations. All regressions were thus performed on the imputed datasets ($N$ = 9,013). However, the available case analyses did not differ substantially. Descriptive statistics are based on the original data.

**2.3.3 Survival analysis.** We used a competing risks model with all-cause death as the competing risk, i.e., as fatal bleeding is already included in BARC 5 [11], this refers to non-bleeding related deaths. The cumulative incidence of major bleeding was estimated by the non-parametric cumulative incidence function, and we report the cumulative incidence at 30 days, 9 months, 12 months, and at 6 years. We also report the incidence rate per 1,000 person-years.

The associations between the explanatory variables and major bleeding were analyzed by cause-specific, standard Cox proportional hazard (PH) regression. After inspection of scaled Schoenfield residuals and graphs in the original dataset and in four of the imputed datasets, we found strong evidence of violation of the PH assumption for acute coronary syndrome.

**Table 1. Descriptive statistics of patients at baseline.** Original data, i.e., available case analysis.

| Variable | | *n* (%) |
|---|---|---|
| Age | < 60 | 3,438 (38.1) |
| | 60–79 | 5,044 (56.0) |
| | 80+ | 531 (5.9) |
| Sex | Male | 6,757 (75.0) |
| Low bodyweight (< 60 kg) | - | 381 (4.3) |
| | *Missing* | 172 |
| Chronic kidney disease | - | 790 (9.2) |
| | *Missing* | 423 |
| Diabetes mellitus | - | 1,123 (12.5) |
| | *Missing* | 25 |
| Hypertension | - | 3,791 (42.4) |
| | *Missing* | 67 |
| Hyperlipidemia | - | 4,868 (54.8) |
| | *Missing* | 135 |
| Current, daily smoking | - | 3,147 (38.1) |
| | *Missing* | 758 |
| Prior CABG | - | 593 (6.6) |
| Prior stroke | - | 346 (3.8) |
| | *Missing* | 14 |
| Prior MI | - | 912 (10.2) |
| | *Missing* | 45 |
| Current ACS | - | 6,319 (70.2) |
| | *Missing* | 14 |
| Multivessel disease | - | 3,574 (39.7) |
| | *Missing* | 5 |
| Stent length | < 20 mm[a] | 4,002 (44.4) |
| | 20–40 mm | 3,311 (36.7) |
| | ≥ 40 mm | 1,700 (18.9) |
| Stent diameter | < 3 mm[a] | 2,945 (32.7) |
| | 3–4 mm | 5,437 (60.3) |
| | ≥ 4 mm | 630 (7.0) |
| | *Missing* | 1 |
| Number of stents | 1[a] | 5,251 (58.3) |
| | 2 | 2,374 (26.3) |
| | ≥ 3 | 1,388 (15.4) |

ACS: Acute coronary syndrome, CABG: Coronary artery bypass graft, CKD: Chronic kidney disease, MI: Myocardial infarction.

[a] 70 patients did not receive a stent after randomization and are included in this group.

We used the modeling strategy and hierarchical backward elimination for interactions as outlined by Kleinbaum and Klein [15]. Variables for the regression model were included based on the literature, biological plausibility, availability and directed acyclic graphs (DAGs). Correlations and unadjusted and adjusted associations informed the DAG for the regression model (S1 Fig in S1 Appendix). The DAG was created using DAGitty 2.3 (www.dagitty.net) [16]. We estimated unadjusted and multivariable adjusted HRs for all included variables; there were no clear indications of multicollinearity. *A priori* (age x sex, weight x sex) and *post hoc* (CKD x diabetes mellitus) interaction terms were assessed by the Wald test.

We assessed the risk of major bleeding among patients suffering a myocardial infarction or undergoing a repeat revascularization during follow-up. A failure variable was created that combined spontaneous MI, periprocedural MI, target lesion PCI, or PCI of new lesion in target vessel or non-target vessel. The MI/PCI survival time was set to the minimum survival time of those endpoints, i.e., time to the *first* event. The major bleeding survival time was split in the case of a MI or PCI, and the resulting time varying indicator variable was included in Cox PH models.

All analyses were performed in Stata 15.1–16.1. A two-side *p* value <0.05 was considered statistically significant. We report 95% confidence intervals, unadjusted for multiple comparisons.

### 2.4 Ethics

NORSTENT (ClinicalTrials.gov: NCT00811772) has been approved by the Regional Committee for Medical and Health Research Ethics (Regional komité for medisinsk og helsefaglig forskningsetikk (REK) nord, 40/2008). Participants provided written informed consent. This substudy is a part of a larger research project, approved by the local Data Protection Officer at the University Hospital of North Norway.

## 3 Results

### 3.1 Baseline characteristics

Baseline characteristics are shown in Table 1. Of the study population, 4509 participants were randomized to BMS and 4504 participants to DES. The study population was predominantly male (75%) with an average age of 63 (± 11) years. The average weight was 83 (± 15) kilograms and average BMI 27 (± 4) kg/m$^2$. Nine percent of patients had chronic kidney disease, defined solely based on eGFR. The prevalence of pharmacologically treated diabetes mellitus, hypertension, or hyperlipidemia were 12%, 42%, and 55%, respectively. Thirty-eight percent of the study population were current daily smokers. The indication for the index PCI procedure was ACS for 70%, stable coronary artery disease for 29%, and a few patients had ischemic heart failure or hemodynamic instability (0.5%). The majority (58%) had one stent implanted, while multivessel disease occurred among 40% of the patients at baseline.

### 3.2 Incidence of major bleeding

In the course of the 42,716 person-years (median time at risk: 4.9 years), 454 events of major bleeding occurred. The six-year cumulative incidence of major bleeding was 5.4% (Fig 1). Sixteen percent (*n* = 74) of the major bleeding cases occurred within the first 30 days after the index PCI (cumulative incidence at 30 days = 0.8%) (Fig 1); half of those cases occurred at the index date or the day after (*n* = 38). The 12-month cumulative incidence was 2.3% (*n* = 209).

The incidence *rate* of major bleeding was 10.6 per 1,000 person-years overall and was highest during the first six-month-interval of follow-up, while being more stable after the first year (Fig 2). The absolute risks, i.e., crude incidence rates, of major bleeding are shown in Table 2 for selected patient sub groups. The highest absolute rates of major bleeding occurred among patients ≥ 80 years (27.2 per 1,000 person-years), among those with chronic kidney disease (25.6 per 1,000 person-years), and among patients with low bodyweight (23.7 per 1,000 person-years). The incidence rates of major bleeding among those randomized to BMS or DES were 10.7 per 1,000 person-years and 10.6 per 1,000 person-years, respectively.

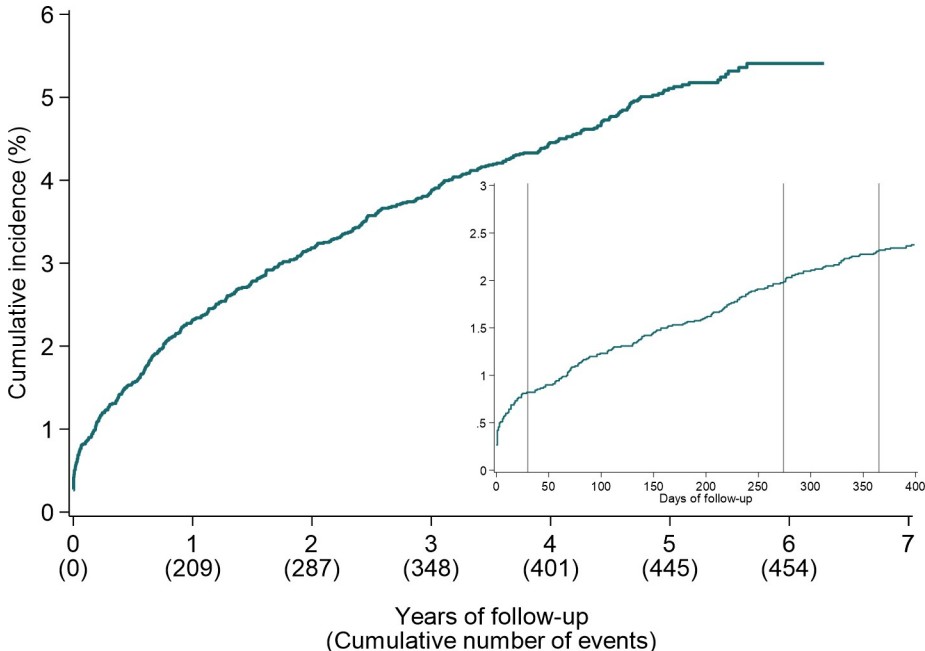

**Fig 1. Cumulative incidence of major bleeding as defined as Bleeding Academic Research Consortium 3–5 according to the non-parametric cumulative incidence function.** Vertical lines in the magnified portion highlight the cumulative incidence at 30 days, 9 months, and 12 months.

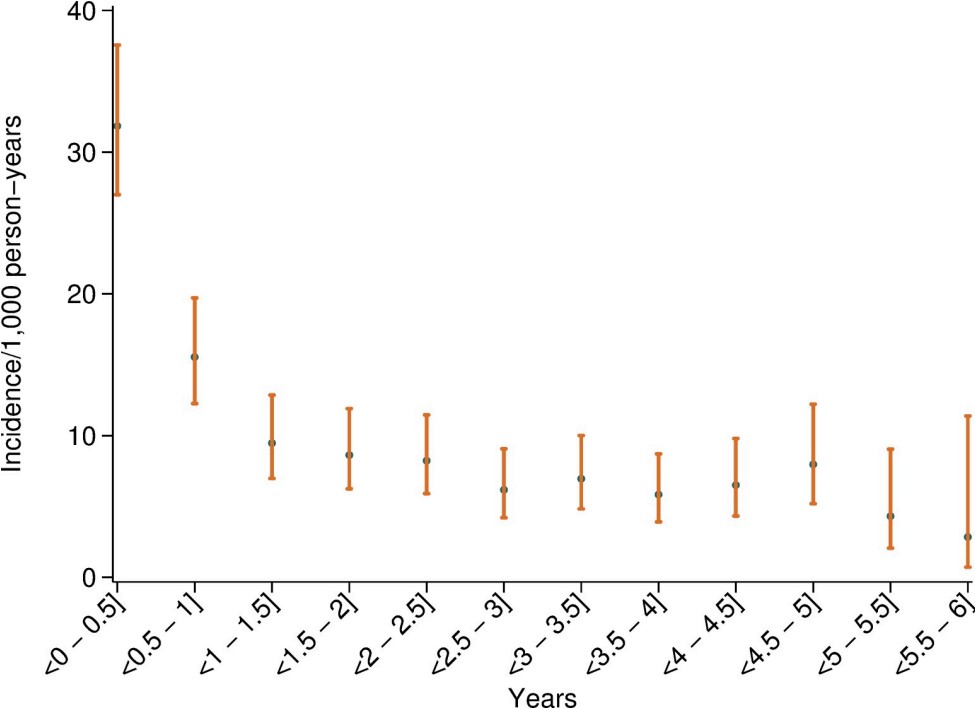

**Fig 2. Incidence rate per 1,000 person-years of major bleeding in consecutive 6-month intervals of the follow-up-period.**

**Table 2. Absolute risks of major bleeding in selected patient sub groups (crude incidence rate per 1,000 patient-years, available case analysis).**

| Variable | Incidence rate (95% CI) |
|---|---|
| *Overall* | *10.6 (9.7–11.7)* |
| Age 80+ | 27.2 (21.1–35.0) |
| Chronic kidney disease | 25.6 (20.8–31.6) |
| Low bodyweight (< 60 kg) | 23.7 (17.3–32.4) |
| Prior stroke | 18.9 (13.2–27.0) |
| Diabetes mellitus | 16.8 (13.6–20.7) |
| Prior CABG | 16.1 (12.0–21.7) |
| Female sex | 14.9 (12.8–17.5) |
| Stent length $\geq$ 40 mm | 13.0 (10.7–15.7) |
| Number of stents $\geq$ 3 | 12.8 (10.3–15.9) |
| Multivessel disease | 12.7 (11.1–14.5) |
| Hypertension | 12.7 (11.1–14.5) |
| Stent diameter < 3 mm | 12.0 (10.3–14.0) |
| Prior MI | 11.9 (9.0–15.7) |
| Current ACS | 10.6 (9.5–11.8) |
| Hyperlipidemia | 10.4 (9.1–11.8) |

ACS: Acute coronary syndrome, CABG: Coronary artery bypass graft, CKD: Chronic kidney disease, MI: Myocardial infarction.

## 3.3 Risk factors

**3.3.1 Unadjusted associations.** The associations of the explanatory variables with major bleeding are shown in Table 3. In unadjusted Cox PH models, chronic kidney disease, low bodyweight (i.e., < 60 kg), female sex, diabetes mellitus, hypertension, prior coronary artery bypass graft, prior stroke, and multivessel disease were statistically significant explanatory variables. Stent properties (total length, diameter, or quantity) were not clearly associated with increased bleeding risk. There were no statistically significant differences in bleeding risk among those *randomized* to BMS compared to DES for the patient subgroups included in this study (S2 Fig in S1 Appendix).

**3.3.2 Multivariable adjusted associations.** In the multivariable adjusted regression model (Table 3), CKD, diabetes mellitus, and low bodyweight were statistically significant risk factors. There was no statistically significant difference in the bleeding risk between men and women, although the point estimate suggested a reduced risk among men (HR = 0.82, 95% CI 0.67–1.03 among men). The bleeding risk increased with age, with three times increased risk among those aged 80 years or older compared to those below 60 years of age.

Having low bodyweight increased the risk of major bleeding by 89% (HR = 1.89, 95% CI 1.32–2.69). Underweight defined as BMI < 18.5 was statistically non-significant but with a similar point estimate (available case analysis; HR = 1.69, 95% CI 0.75–3.83). Weight showed a non-linear association with bleeding risk. The interaction tests with age ($p$ = 0.555) or sex ($p$ = 0.282) were non-significant, although stratification by sex suggested a trend towards an increase particularly among women (S1 Table in S**1** Appendix).

Diabetes mellitus increased the risk of major bleeding by 56% (HR = 1.56, 95%CI 1.22–2.00). The sensitivity analysis, with diabetes mellitus being defined as HbA1C $\geq$ 6.5%, showed a similar point estimate (available case analysis; HR = 1.38, 95% CI 0.96–2.00, $n$ = 4,364,).

Chronic kidney disease increased the risk of major bleeding by 77% (HR = 1.77, 95% CI 1.37–2.29). There was no statistically significant multiplicative interaction between CKD and diabetes mellitus (*post hoc* analysis, $p$ = 0.359).

**Table 3. Hazard ratios for major bleeding in unadjusted and mutually adjusted models.** The dataset has been imputed to $N = 9,013$ by multiple imputation.

| | | Unadjusted | | Adjusted[a] | |
|---|---|---|---|---|---|
| | | HR | 95% CI | HR | 95% CI |
| CKD | No CKD | 1 | Ref | 1 | Ref |
| | CKD | **2.65** | (2.10–3.35) | **1.77** | (1.37–2.29) |
| Hypertension | No hypertension | 1 | Ref | 1 | Ref |
| | Hypertension | **1.39** | (1.16–1.67) | 1.12 | (0.92–1.38) |
| Diabetes mellitus | No diabetes | 1 | Ref | 1 | Ref |
| | Diabetes | **1.69** | (1.33–2.14) | **1.56** | (1.22–2.00) |
| CABG | No prior CABG | 1 | Ref | 1 | Ref |
| | Prior CABG | **1.54** | (1.12–2.11) | 1.25 | (0.87–1.79) |
| Stroke | No prior stroke | 1 | Ref | 1 | Ref |
| | Prior stroke | **1.82** | (1.26–2.63) | 1.39 | (0.95–2.02) |
| Weight | ≥ 60 kg | 1 | Ref | 1 | Ref |
| | < 60 kg | **2.44** | (1.76–3.38) | **1.89** | (1.32–2.69) |
| Age | < 60 | 1 | Ref | 1 | Ref |
| | 60–79 | **2.22** | (1.76–2.79) | **1.97** | (1.54–2.52) |
| | 80+ | **4.60** | (3.33–6.35) | **3.03** | (2.09–4.38) |
| Sex | Female | 1 | Ref | 1 | Ref |
| | Male | **0.62** | (0.51–0.76) | 0.82 | (0.67–1.03) |
| Hyperlipidemia | No hyperlipidemia | 1 | Ref | 1 | Ref |
| | Hyperlipidemia | 0.96 | (0.80–1.16) | 0.85 | (0.69–1.06) |
| Prior MI | No prior MI | 1 | Ref | 1 | Ref |
| | Prior MI | 1.14 | (0.85–1.53) | 0.88 | (0.64–1.21) |
| Stent length | < 20 mm | 1 | Ref | 1 | Ref |
| | 20–40 mm | 0.95 | (0.77–1.18) | 1.08 | (0.82–1.44) |
| | ≥ 40 mm | 1.25 | (0.99–1.59) | 1.51 | (0.95–2.40) |
| Stent diameter | < 3 mm | 1 | Ref | 1 | Ref |
| | 3–4 mm | 0.87 | (0.71–1.04) | 0.96 | (0.78–1.18) |
| | ≥ 4 mm | 0.70 | (0.46–1.06) | 0.90 | (0.58–1.40) |
| Number of stents | 1 | 1 | Ref | 1 | Ref |
| | 2 | 0.99 | (0.80–1.24) | 0.84 | (0.60–1.16) |
| | ≥ 3 | 1.24 | (0.97–1.59) | 0.79 | (0.49–1.27) |
| Current ACS | No ACS | - | - | 1 | Ref |
| | ACS | | | | * |
| Multivessel disease | No multivessel disease | 1 | Ref | 1 | Ref |
| | Multivessel disease | **1.34** | (1.11–1.61) | 1.08 | (0.87–1.35) |
| Current, daily smoking | No current smoking | 1 | Ref | 1 | Ref |
| | Current smoking | 0.89 | (0.74–1.09) | 1.19 | (0.95–1.47) |

[a] Mutually adjusted for all listed variables as well as PCI hospital.

* Violation of PH assumption. Modeled as a time varying covariate in the complete case dataset, $t_0$: $HR_{t0}$ = 1.43 (95% CI 1.00–2.05), $HR_{interaction}$ = 0.85 (95% CI 0.74–0.98), and as a regular variable in the imputation model. ACS: Acute coronary syndrome, CABG: Coronary artery bypass graft, CKD: Chronic kidney disease, MI: Myocardial infarction.

**3.3.3 Myocardial infarction or revascularization during follow-up.** Patients with MI/PCI during follow-up were generally older, and more likely to have CKD, diabetes mellitus, hypertension, hyperlipidemia, previous stroke, CABG, or MI, multivessel disease, as well as longer total stent length, lower mean stent diameter, and higher number of stents (S2 Table in

S1 Appendix). The 12-month cumulative incidence among those with MI/PCI during follow-up was 3.9% compared to 2.3% in those without (Kaplan-Meier estimate). MI/PCI during follow-up increased the risk of major bleeding by 67% (HR = 1.67, 95% CI 1.29–2.15), after adjustment (S3 Table in S1 Appendix).

### 3.4 Adherence

Adherence to antiplatelet therapy was assessed through regular interviews, and these results can be found in the supplement (S4 Table in S**1** Appendix).

## 4 Discussion

### 4.1 Main findings

In this large, pragmatic randomized trial on PCI with stent implantation and subsequent DAPT among stent naïve patients, we found a 12-month cumulative incidence of BARC major bleeding of 2.3%. The presence of chronic kidney disease, diabetes mellitus, low body-weight, or advanced age increased the risk of major bleeding. A myocardial infarction or a repeat revascularization during follow-up also increased the risk of major bleeding.

### 4.2 Incidence of major bleeding

The incidence of major bleeding seems higher than reported in previous RCTs, particularly explanatory RCTs, which included selected patient populations with lower bleeding risk [17]. Investigators of the SECURITY RCT report a 12-month cumulative incidence of BARC 3 or 5 (i.e., CABG related bleeding excluded) major bleeding after DAPT with clopidogrel of 1.1% [18]. Other explanatory RCTs report a 12-month cumulative incidence in the range 0.5–1.5%, with differing bleeding definitions [2,17], while the 12-month cumulative incidence of Thrombolysis in Myocardial Infarction (TIMI) major bleeding was around 0.7% (1.3% if minor bleeding is included) in the derivation cohort for the PRECISE-DAPT score; bleeding events in the first seven days were excluded [3]. In the DAPT RCT, 2.7% of patients were excluded before randomization, due to a GUSTO major bleeding during the initial 12 months [19]. In contrast, in all-comer RCTs, such as PRODIGY and GLOBAL LEADERS, which included broader patient populations, the 12-month cumulative incidence of BARC 3 or 5 bleeding were 2.0%, and 1.9%, respectively [17], more similar to our findings.

A lower incidence is reported in some registry-based studies. In a recent registry-based validation study of the DAPT score, major bleeding occurred among 1.2% during the first 12 months of DAPT with mainly clopidogrel [9]. In the EPICOR registry, among patients with ACS, the two-year cumulative incidence of bleeding was 3.6%, where half (1.7%) were deemed clinically significant [20].

A higher incidence has been found in other registry-based studies. The finding of a 12-month cumulative incidence of BARC major bleeding of 2.8% from the PARIS registry [21] is noteworthy. In Danish registry studies, DAPT with clopidogrel after MI or atrial fibrillation was associated with a yearly incidence of bleeding of 3.7% [22] or 5.0% [23], respectively.

In summary, the cumulative incidence of major bleeding in NORSTENT is seemingly higher than in previous explanatory RCTs, more comparable to other all-comers RCTs, and lower than in some, but not all, registry-based studies. A direct comparison is challenging, particularly due to varying bleeding definitions. However, the highly selected, explanatory RCTs include a higher proportion of low-risk patients, while pragmatic RCTs and registry-based studies might be closer to real-life practice. For example, patients with CKD or uncontrolled hypertension were excluded from SECURITY [18].

### 4.3 Risk factors for major bleeding

We identified chronic kidney disease, diabetes mellitus, low bodyweight, and older age as independent risk factors for major bleeding. Stent properties were, as expected, not apparently associated with the risk of major bleeding. We have previously shown that stent type (DES or BMS) is not associated with major bleeding [10]. Since the study inclusion period (2008–11), there has been a reduction in the use of BMS; the rate of use of BMS was below one percent in 2017 in countries like Norway [24, p. 28] or Italy [25]. As the DAPT duration historically was shorter for BMS, which resulted in a reduced bleeding risk, BMS was preferentially used among patients with high bleeding risk. However, this trend did not influence the findings in this study, since the participants had similar (or close to similar) DAPT duration regardless of stent type. Chronic kidney disease was among the strongest risk factors, both in absolute and relative terms, for major bleeding. Chronic kidney disease is an established risk factor for bleeding [17], possibly due to reduced platelet reactivity [26], and creatinine clearance is incorporated into several risk prediction tools, e.g., the PRECISE-DAPT score [3].

Diabetes mellitus is a well-established ischemic risk factor, although guidelines state that the presence of diabetes mellitus should not affect DAPT duration [1]. The potential role of diabetes mellitus as a risk factor for bleeding is more controversial. In the derivation cohort of the PRECISE-DAPT score, insulin-dependent diabetes mellitus was associated with a 65% increased risk of TIMI major or minor bleeding in unadjusted analyses [3]; diabetes mellitus is, however, not included in the final PRECISE-DAPT score. While in the joint ischemic and bleeding risk "DAPT score", diabetes mellitus adds to the ischemic side of the equation [4].

Lemesle et al. recently reported findings from the CORONOR registry of an increased risk of bleeding in patients with stable coronary artery disease and diabetes mellitus, treated with aspirin and/or clopidogrel (roughly 40% received clopidogrel; HR = 1.75, 95% CI 1.05–2.91 among diabetic patients with HbA1C level $\geq$ 7%) [27]. Simek et al., on the contrary, recently report no association between diabetes mellitus and bleeding in MI patients treated with prasugrel or ticagrelor, based on information from the PRAGUE-18 trial [28]. The authors note that the bleeding risk among diabetic patients may be differential among users of clopidogrel and, particularly, prasugrel [28]. The patients included in the CORONOR, NORSTENT, and the aforementioned derivation cohort of the PRECISE-DAPT score, where an association between diabetes mellitus and bleeding has been found, mainly received clopidogrel and ASA. The exact mechanisms whereby diabetes mellitus is associated with bleeding risk is unknown. Careful interpretation is advisable while we await further data on the association between diabetes mellitus, DAPT, and bleeding risk.

Patients weighing less than 60 kg, predominantly women, were at increased risk of major bleeding. In the TRITON-TIMI 38 RCT, a body weight < 60 kg was found to be a risk factor for serious bleeding in an ACS population (HR = 2.30, 95% CI 1.74–3.05) [29]. One explanation could be the increased systemic availability of ASA or other antithrombotic drugs in patients with low bodyweight; Rothwell et al. report increased risk of major bleeding only among ASA users with lower body weight (90-kilogram threshold) [30]. Being underweight may also act as a proxy variable for frailty. Frailty is associated with increased bleeding risk after PCI [31], but whether this is causal is controversial [17]. The pre-specified cut-off at < 60 kg was somewhat arbitrarily chosen, but based on literature [29].

Patients who suffered an MI or PCI during follow-up had a 67% increased risk of major bleeding. Bleeding is a common complication of PCI, while an MI or PCI would normally result in a new or extended course of DAPT, increasing the risk of bleeding. As the study period covered the time period of the introduction of prasugrel and ticagrelor in contemporary practice in Norway (~2010) [32], patients with an MI during follow-up would be more

likely to receive one of those, with a corresponding increased bleeding risk [1]. Finally, as patients suffering a recurrent MI generally perform worse, and have more comorbidities and risk factors [33], the finding could, partially, be due to residual confounding.

### 4.4 Strengths

NORSTENT is a researcher-initiated, pragmatic RCT [34], funded by not-for-profit organizations, that recruited more than 70% of the eligible population and nearly 50% of all patients undergoing PCI in Norway during the inclusion period. The study has a long follow-up period, which was complete for all patients by using mandatory national health registries covering the entire population, giving insight into the risk of bleeding in the PCI population beyond the period of DAPT. The bleeding endpoints were clinically adjudicated by a blinded endpoint committee, increasing the validity. The sample size, as opposed to several of the previous RCTs [17], allowed detection of smaller effect sizes (e.g., HR $\approx$ 1.3–1.5).

### 4.5 Weaknesses

We lacked information on other risk factors, e.g., anemia, low white blood cell count, use of NSAIDs, and frailty, although age, sex, and weight could to some degree capture the latter [31]. Furthermore, patients with a contraindication against DAPT, e.g., history of bleeding or thrombocytopenia, users of anticoagulants, and patients with known cancer were excluded, leading to a possible underestimation of the bleeding risk compared to contemporary clinical practice.

Since the patients were prescribed clopidogrel for at least nine months according to the protocol, and the introduction of prasugrel and ticagrelor happened midway through the study period, we were unable to compare different P2Y12 inhibitors or durations of DAPT in this analysis. Finally, current PCI procedural techniques have some important differences from 2008–11, including lower use of glycoprotein IIb/IIIa Inhibitors, and higher frequency of radial access site, resulting in a lower risk of bleeding [35,36].

## 5 Conclusion

In this large, pragmatic RCT, among patients randomized to PCI with BMS or DES, the 12-month cumulative incidence of major bleeding was 2.3%. This is twice as high as in previous explanatory RCTs that included more selected patient populations with lower bleeding risk. Independent risk factor for major bleeding were chronic kidney disease, diabetes mellitus, low bodyweight, and older age, while an MI or PCI during follow-up increased the risk of major bleeding by 67%. Diabetes mellitus is well established as an ischemic risk factor, while its role as a bleeding risk factor warrants further research.

## Supporting information

**S1 Appendix.**
(DOCX)

## Acknowledgments

We thank the investigators, clinicians, and participants involved in NORSTENT for their invaluable contribution, and colleagues at RELIS, the Epidemiology of Chronic Diseases research group, and Prof. Rune Wiseth for feedback to this manuscript.

## Author Contributions

**Conceptualization:** Per-Jostein Samuelsen, Anne Elise Eggen, Terje Steigen, Kaare Harald Bønaa.

**Data curation:** Tom Wilsgaard, Kaare Harald Bønaa.

**Formal analysis:** Per-Jostein Samuelsen, Tom Wilsgaard.

**Funding acquisition:** Per-Jostein Samuelsen, Anne Elise Eggen, Terje Steigen, Tom Wilsgaard, Kaare Harald Bønaa.

**Investigation:** Terje Steigen, Andreas Kristensen, Anne Skogsholm, Elizabeth Holme, Christian van den Heuvel, Bjørn Bendz, Dennis W. T. Nilsen, Kaare Harald Bønaa.

**Methodology:** Per-Jostein Samuelsen, Anne Elise Eggen, Terje Steigen, Tom Wilsgaard, Kaare Harald Bønaa.

**Resources:** Kaare Harald Bønaa.

**Software:** Per-Jostein Samuelsen.

**Supervision:** Kaare Harald Bønaa.

**Validation:** Kaare Harald Bønaa.

**Visualization:** Per-Jostein Samuelsen.

**Writing – original draft:** Per-Jostein Samuelsen.

**Writing – review & editing:** Per-Jostein Samuelsen, Anne Elise Eggen, Terje Steigen, Tom Wilsgaard, Andreas Kristensen, Anne Skogsholm, Elizabeth Holme, Christian van den Heuvel, Jan Erik Nordrehaug, Bjørn Bendz, Dennis W. T. Nilsen, Kaare Harald Bønaa.

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
