## [Decision Letter · Decision Letter 0]

23 Jul 2020

PONE-D-20-14185

Incidence and risk factors for major bleeding among patients undergoing percutaneous coronary intervention: findings from the Norwegian Coronary Stent Trial (NORSTENT).

PLOS ONE

Dear Dr. Samuelsen,

Thank you for submitting your manuscript to PLOS ONE. After careful consideration, we feel that it has merit but does not fully meet PLOS ONE’s publication criteria as it currently stands. Therefore, we invite you to submit a revised version of the manuscript that addresses the points raised during the review process.

We look forward to receiving your revised manuscript.

Kind regards,

Salvatore De Rosa

Academic Editor

PLOS ONE

Journal Requirements:

2.We note that you have indicated that data from this study are available upon request. PLOS only allows data to be available upon request if there are legal or ethical restrictions on sharing data publicly. For information on unacceptable data access restrictions, please see http://journals.plos.org/plosone/s/data-availability#loc-unacceptable-data-access-restrictions.

4.Thank you for including your ethics statement:  'NORSTENT (ClinicalTrials.gov: NCT0081177) has been approved by the Regional Ethics Committee

118 (REK Nord 40/2008)'.    

(a) Please amend your current ethics statement to include the full name of the ethics committee/institutional review board(s) that approved your specific study.  

(b) Once you have amended this/these statement(s) in the Methods section of the manuscript, please add the same text to the “Ethics Statement” field of the submission form (via “Edit Submission”).

5. Please ensure you have included the correct clinicaltrials.gov registration number for the clinical trial referenced in the manuscript. Currently, the clinicialtrials.gov provided in the manuscript is NCT0081177, whereas the correct registration numbers appears to be NCT00811772.

Additional Editor Comments (if provided):

The authors report original results on predictors of bleeding events in the large NORSTENT cohort.

Data analysis shows no impact by the type of stent implantes (DES vs BMS) on bleeding. This is an important confirmation of previous results in a large study cohort that fit in the lane of previous evidence. A recent nationwide report (Giannini F. et al. Italian Multicenter Registry of Bare Metal Stent Use in Modern Percutaneous Coronary Intervention Era (AMARCORD): A multicenter observational study. Catheter Cardiovasc Interv. 2020 Mar 21. doi: 10.1002/ccd.28798. Online ahead of print.) showed a hyperbolic reducion in BMS use from 2013 to rates lower than 1% in 2016). This phenomenon depicts a trend that was sustanined by a growing body of evidence on the use of DES in different clinical context, including high bleeding risk patients. Please discuss this aspect.

Reviewers' comments:

Reviewer's Responses to Questions

**Comments to the Author**

1. Is the manuscript technically sound, and do the data support the conclusions?

Reviewer #1: Yes

Reviewer #2: Yes

2. Has the statistical analysis been performed appropriately and rigorously? 

Reviewer #1: Yes

Reviewer #2: Yes

3. Have the authors made all data underlying the findings in their manuscript fully available?

Reviewer #1: No

Reviewer #2: Yes

4. Is the manuscript presented in an intelligible fashion and written in standard English?

Reviewer #1: Yes

Reviewer #2: Yes

5. Review Comments to the Author

Reviewer #1: In this manuscript, Samuelsen investigated potential risk factors of major bleeding in the Norwegian Coronary Stent Trial. More than 9000 patients were included into the randomized control trial and followed for a median of five years. The authors observed a cumulative incidence rate of 2.3%. Additional risk factors were also identified to relate to major bleeding such as CKD, underweight, diabetes and advanced age. Overall, the study is interesting and the manuscript was well written.

Following are some comments to be considered.

• The authors merged both study arms. However, it is unclear how many samples were assigned to each study arm. It would also be useful to compare the risk factors derived from either arm.

• In addition to MI, what other ischemic events were studied?

• Is there any control for medication?

• The authors indicated that “the combined proportion of missing observations was 17% with a missing at random pattern”. How the missing at random pattern was assessed?

• Is there any normalization of the study variables?

• How the baseline was defined for patients with multiple stent implanted? Was it counted since the last stent was implanted?

• Is there any bleeding prior to the stent implantation?

• In figure 1, please list the cumulative number of events under each year of follow-up

Reviewer #2: In the manuscript entitled "Incidence and risk factors for major bleeding among patients undergoing percutaneous coronary intervention: findings from the Norwegian Coronary Stent Trial (NORSTENT) " authors lead by Per-Jostein Samuelsen investigated the major bleeding risk factors in their cohort of 9013 patients.

This analysis shows chronic kidney disease (CKD), advanced age, and underweight as risk factors for major bleeding among patients receiving DAPT after PCI. Moreover, diabetes mellitus or recurrent MI increase bleeding risk during the follow-up.

The authors should be commended in their effort, as this analysis could support prior reports giving insights on this relevant problem. This reviewer has the following comments to raise with respect to the manuscript:

• Impact of the effective duration of DAPT is not presented in the document. Could the authors present this data, if available, and put it into perspective towards the effect of the various predictors? Is longer DAPT duration a predictor for bleeding in this analysis, could the author estimate the increase in bleeding each month of extended DAPT above 9 months?

• PCI or MI during follow-up increased risk of bleeding. This is probably due to extending DAPT after additional procedures and subsequent peri-procedural/intrahospital events. Was this adjusted for overall DAPT duration? If this is not the case this is not interesting and the text dedicated to this result shortened.

• BARC bleeding 3,4 or 5. Why the authors decided to also include BARC 4 in the primary endpoint ? This is hardly a marker of out-of hospital bleeding related to DAPT.

• It would be meaningful to present information on the bleeding sites.

• Was a treatment with clopidogrel made for all patients ? Or novel P2Y12 inhibitors have been also used ? If this is not the case this should be added in the limitation section.

• Could the authors validate PRECISE-DAPT score in their population? A validation of the DAPT score would not be feasible since most patients did not do more than 9 months of DAPT.

• Discussion section, for the PRECISE-DAPT please report the incidence of TIMI Major or minor bleeding, comparison between BARC 3 or 5 and TIMI major bleeding only would be misleading.

• The higher reported event rate could be also due to procedural characteristics than in 2008 -2011 were much different than today (e.g. bulkier access site, higher use of GPI). This should be reported in the study discussion and limitations.

• Discussion, diabetes. "This could lead to diabetic ulcerations of the feet, and retinopathy. 257 The latter could cause intraocular bleeding compromising vision, which is classified as a BARC major 258 bleeding [8]." Could the authors support this statement with prior data from the literature. If this is not the case the conclusion here seems farfetched. In addition, the rate of intraocular bleeding is very low and hardly explain alone the association of diabetes with bleeding.

• Abstract, row 7: the authors repeated the word “randomized” twice.

• Line 31: please state international guidelines: also Canadian guidelines and Japanese guidelines support standardized risk scores for DAPT.

• While citing Rothwell et al. wasn't the threshold used in that manuscript 70kg rather than 90 kg ?

• Supplementary Figure 1 is interesting. DAG is very informative and graphic. Could the authors please explain why older age, that is independently associated to major bleeding, also provide an effect through diabetes and CKD ? is this a clinical assumption or the product of analysis ?

• Figure 2 is really informative. However, I believe that the higher rates of bleeding presented in the first block (0 to 6 months) are driven by intrahospital/periprocedural bleeding (especially considering the practice in 2008-2011 in which femoral access and GPIIb/IIIa were more extensively used). Excluding those events from the analysis would be much more informative to provide a snapshot of the bleeding risk during therapy during out-of-hospital follow-up.

• Table 3 could be simplified shortened eliminating the reference lines. Information about the reference groups could be included in the rows or in the footnotes.

• Table 3 there is a typo for stent length >40mm (1.5q).

6. PLOS authors have the option to publish the peer review history of their article (what does this mean?). If published, this will include your full peer review and any attached files.

Reviewer #1: No

Reviewer #2: No

---

## [Author Response · Author response to Decision Letter 0]

30 Dec 2020

The response to the reviewer and editor comments is attached ("Response to Reviewers")

---

## [Decision Letter · Decision Letter 1]

8 Feb 2021

Incidence and risk factors for major bleeding among patients undergoing percutaneous coronary intervention: findings from the Norwegian Coronary Stent Trial (NORSTENT)

PONE-D-20-14185R1

Dear Dr. Samuelsen,

We’re pleased to inform you that your manuscript has been judged scientifically suitable for publication and will be formally accepted for publication once it meets all outstanding technical requirements.

Kind regards,

Salvatore De Rosa

Academic Editor

PLOS ONE

Additional Editor Comments (optional):

Reviewers' comments:

Reviewer's Responses to Questions

**Comments to the Author**

1. If the authors have adequately addressed your comments raised in a previous round of review and you feel that this manuscript is now acceptable for publication, you may indicate that here to bypass the “Comments to the Author” section, enter your conflict of interest statement in the “Confidential to Editor” section, and submit your "Accept" recommendation.

Reviewer #1: All comments have been addressed

2. Is the manuscript technically sound, and do the data support the conclusions?

Reviewer #1: Yes

3. Has the statistical analysis been performed appropriately and rigorously? 

Reviewer #1: Yes

4. Have the authors made all data underlying the findings in their manuscript fully available?

Reviewer #1: Yes

5. Is the manuscript presented in an intelligible fashion and written in standard English?

Reviewer #1: Yes

6. Review Comments to the Author

Reviewer #1: (No Response)

7. PLOS authors have the option to publish the peer review history of their article (what does this mean?). If published, this will include your full peer review and any attached files.

Reviewer #1: No

---

## [Editor Report · Acceptance letter]

17 Feb 2021

PONE-D-20-14185R1 

Incidence and risk factors for major bleeding among patients undergoing percutaneous coronary intervention: findings from the Norwegian Coronary Stent Trial (NORSTENT). 

Dear Dr. Samuelsen:

I'm pleased to inform you that your manuscript has been deemed suitable for publication in PLOS ONE. Congratulations! Your manuscript is now with our production department. 

Kind regards, 

on behalf of

Dr. Salvatore De Rosa 

Academic Editor

PLOS ONE